# Seismic Risk Assessment of Typical Reinforced Concrete Frame School Buildings in Sri Lanka

**Tharindu Malinga Abeysiriwardena** [1], **Kushan Kalmith Wijesundara** [1] **and Roberto Nascimbene** [2,*]

1   Department of Civil Engineering, University of Peradeniya, Peradeniya 20400, Sri Lanka;
    malingathari@gmail.com (T.M.A.); kushanw@eng.pdn.ac.lk (K.K.W.)
2   Istituto Universitario di Studi Superiori (IUSS), Scuola Universitaria Superiore (IUSS), 27100 Pavia, Italy
*   Correspondence: roberto.nascimbene@iusspavia.it

**Abstract:** The assessment of seismic risk for critical and strategic structures like schools and hospitals remains crucial, even in regions with low seismic activity. Presently, operational school buildings in Sri Lanka are primarily designed to handle gravitational loads without considering capacity-based design principles. Consequently, these structures may lack the necessary lateral resistance to mitigate potential damage or collapse during future earthquakes in Sri Lanka. Hence, conducting seismic risk assessments for such school buildings is imperative to ensure the safety of their occupants. In this research paper, we utilize a recently developed probabilistic seismic hazard map for Sri Lanka to evaluate seismic risk. We employ two nonlinear 3-D finite element models of school buildings created in OpenSees. Incremental Dynamic Analysis is conducted using a well-established set of ground motions, continuing until the structure approaches the point of collapse, to determine the probability of collapse prevention. Subsequently, we develop fragility functions for two limit states, immediate occupancy, and collapse prevention. These fragility curves are then used to compute the probability of exceeding these limit states, aiding in the assessment of the structural safety of the school buildings. A key outcome of this analysis reveals a general trend of increased damage probabilities as the number of stories in the buildings increases despite the distinct structural characteristics of each building. It is also important to note that the disparities between the immediate occupancy and more severe damage cases, such as collapse prevention, are notably pronounced in both two- and three-story school buildings.

**Keywords:** existing buildings; seismic risk; fragility functions; limit state; life safety; collapse prevention; advanced numerical models





## 1. Introduction

Sri Lanka is situated in the north-western region of the Indo-Australian plate, far removed from plate boundaries. Consequently, Sri Lanka is typically regarded as being shielded from inter-plate seismic activities. However, it is important to note the presence of intra-plate activities along the North-West to South-West coast of Sri Lanka, in close proximity to the well-known but failed Mannar rift zone and the Comorin ridge. These intra-plate activities assume significance in the seismic risk assessment of critical and strategic structures in Sri Lanka, particularly schools [1]. In recent times, due to earthquakes occurring in South Asia, notably in countries such as India [2–4], Pakistan [5], and Nepal [6,7], school buildings have endured severe damage. This is primarily because many of these structures were originally designed to withstand only gravitational forces. The complete collapse of such buildings, particularly those constructed as reinforced concrete (RC) frame buildings, tragically resulted in the loss of many young lives among school children. This is not an isolated trend limited to the Eastern part of the world, including Sri Lanka; rather, it is a global phenomenon supported by numerous instances worldwide. Recent major earthquakes in countries like Italy [8,9], New Zealand [10–12], and Japan [13,14] have underscored the vulnerability of existing RC structures that often adhere to outdated building

analysis, design, and construction practices, primarily intended for gravity loads. These instances have brought to light the inadequate structural performance witnessed during recent seismic events across the globe. A key factor contributing to this vulnerability is the ineffective transfer of horizontal forces, particularly between structural elements and ancillary, secondary, or non-structural components, including slabs [15]. On-site investigations and reconnaissance teams have shed light on the significant structural deficiencies of RC buildings and industrial facilities, which were evident in past earthquakes such as the 1998 Adana–Ceyhan [16], 1999 Kocaeli, and Duzce earthquakes [17,18]. These inadequacies encompass issues related to stiffness, ductility, energy dissipation mechanisms, strength, and poorly designed connection details, all of which played a crucial role in the observed problems. Within the European context, the 2009 L'Aquila earthquake served as a wake-up call, highlighting numerous critical issues. This event underscored the urgent necessity of implementing specific capacity design rules, establishing a hierarchy of resistance, and considering various failure modes as additional limit states in classical design methodologies [19,20]. A similar pattern of behavior and consequent collapses was also evident in the 2010 Chilean earthquake, as documented by Ghosh and Cleland [21]. Moreover, just two years after the Chilean earthquake, during the 20th and 29th of May in 2012, the Central-Italy Emilia earthquakes [22,23] once again highlighted the vulnerability of RC structures that typify the non-code-compliant Italian building practices. In many of these structures, substantial uncertainties are associated with their nonlinear behavior. These uncertainties encompass factors such as the existence and precise location of potential plastic hinge zones and their ductility capacity, which are often inadequately established and understood.

The aftermath of these significant earthquakes, marked by extensive and persistent structural damage, underscores the pressing need for dependable and secure methodologies. These methodologies should encompass detailed analyses, advanced modeling techniques, and the enhancement and evaluation of existing structures. They should also account for the intricate interplay among structural components to yield more precise insights into the nonlinear dynamic responses of buildings. This demand becomes even more critical when considering the numerous buildings with strategic or public functions, such as schools, hospitals, or government defense facilities. To the best of our knowledge, there has been limited research conducted on the vulnerability of school buildings in Sri Lanka. Therefore, in light of recent global events and the insights provided in the scientific literature [1–23], this paper aims to employ established methodologies to investigate, analyze, and assess typical RC frame school buildings in Sri Lanka. These findings will emphasize the crucial significance of engineering school buildings to effectively withstand the lateral forces generated by seismic events. Additionally, they will serve as essential inputs for shaping seismic rehabilitation plans and estimating the necessary budget to enhance the safety levels of existing low-ductile RC frames. The next phase of this study could encompass strengthening structural components through various means such as concrete jacketing, steel jacketing, Fiber Reinforced Polymer (FRP) wrapping, the application of pre-stress components, and the implementation of dampers.

In 2014, census data revealed some critical statistics [24–26]. Sri Lanka, with a total population of 20 million, accommodates around 4.1 million students and teachers who frequent government schools for five days each week. This constitutes nearly 20 percent of the entire population in the country. Additionally, a significant portion of school buildings in Sri Lanka conform to a standard typology characterized by two or three stories, and they are primarily designed to withstand gravitational forces alone. Consequently, these school buildings might lack the necessary lateral load resistance capacity, rendering them highly susceptible to seismic hazards, as evidenced by past earthquake events worldwide. Considering all these variables, which encompass the exposure of children during school hours and the vulnerability of their lives in these structures, it becomes paramount and essential to evaluate the seismic risk associated with school buildings in Sri Lanka.

Additionally, this paper leverages a recently created probabilistic seismic hazard map specific to Sri Lanka to assess seismic risk. This seismic hazard map for Sri Lanka incorporates the mean Peak Ground Acceleration (PGA) value at each location. It delineates two seismic zones, as depicted in Figure 1, labeled Zone 1 and Zone 2, with design PGA values of 0.1 g and 0.05 g, respectively.

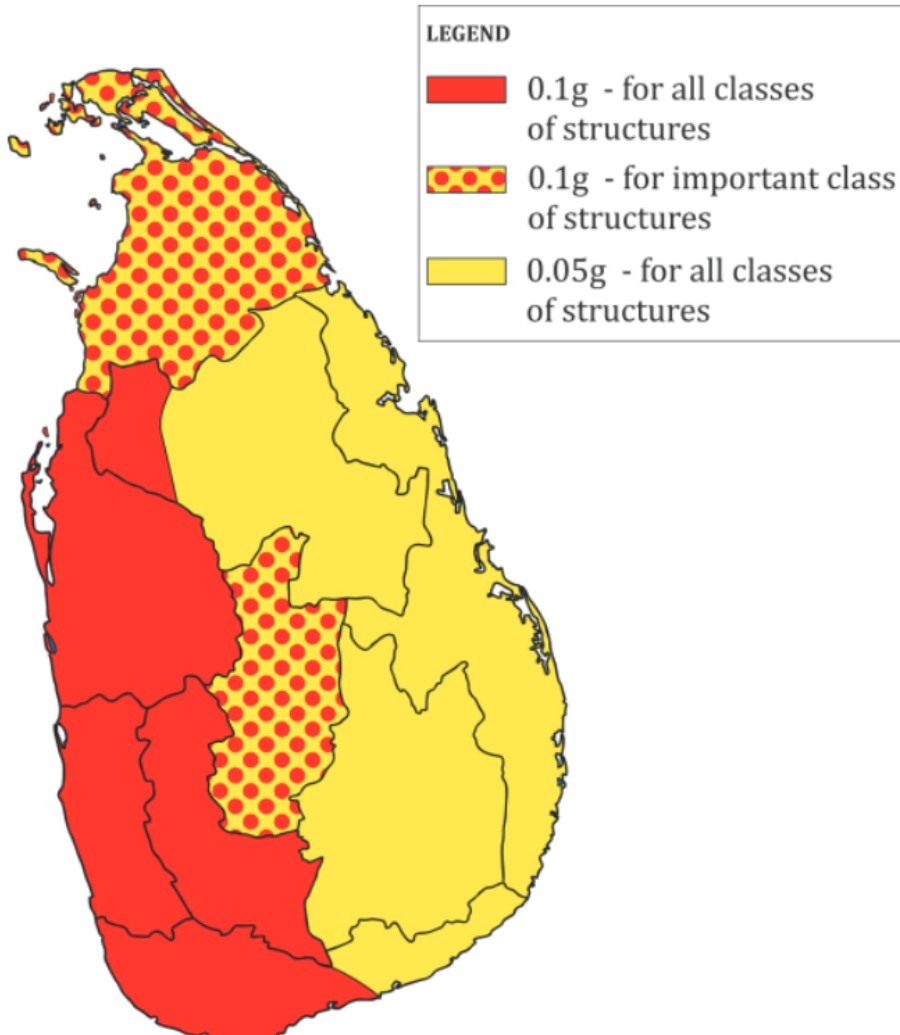

**Figure 1.** Seismic Hazard Map for Sri Lanka based on PGA: Zone 1 and Zone 2 having design PGA values of 0.1 g and 0.05 g.

In the assessment of seismic risk for structures, researchers commonly rely on fragility curves. These curves depict the correlation between the likelihood of damage and the intensity of an earthquake. Essentially, a fragility function quantifies the probability of a structure (or non-structural element) experiencing certain degrees of damage in response to a specific level of ground motion during an earthquake. Seismic fragility, in its fundamental form, can be mathematically expressed and characterized by fragility curves, which outline the probability of reaching or surpassing a predefined limit state (e.g., damage, loss of support, buckling) at a given ground shaking intensity corresponding to a damage index.

Inside the topic of seismic risk assessment, various damage indices have been introduced in previous studies to estimate the limit state a building reaches when subjected to a specific level of earthquake ground motion. Seminal research by Cosenza et al. [27] and Bozorgnia and Bertero [28] have comprehensively documented and reviewed a wide array of damage indices proposed over the years. For this particular study, the inter-story drift damage index has been selected as a more suitable non-modal parameter-based damage

index for quantifying expected building damage. The inter-story drift damage index is defined as the ratio between the maximum inter-story drift at the center of mass and the ultimate inter-story drift, typically corresponding to a 20% strength reduction in the entire story.

The development of fragility curves can be achieved through various methodologies available in the scientific literature. However, it is worth noting that there is no universally accepted and standardized approach to this process, as recognized by the scientific community [29]. In this specific study, fragility curves are formulated based on the median inter-story drift as a damage index, using a deterministic approach. The dispersion of the lognormal cumulative probability distribution function [30] is employed to evaluate the seismic risk of school buildings, focusing on two distinct performance objectives: immediate occupancy (IO) and collapse prevention (CP). The median response and dispersion parameters are determined through Incremental Dynamic Analysis (IDA), as proposed by Vamvatsikos and Cornell [31–33].

Section 2 of this study provides detailed descriptions of the case studies, focusing on a typical two-story school building and a more conventional three-story school building. In Section 3, the emphasis shifts to the numerical models used, while Section 4 delves into the input ground motion data. The subsequent Section 5 centers on the results of the IDA curves. This analysis highlights a noteworthy vulnerability in both the two- and three-story RC school buildings, particularly in the longitudinal direction, as detailed in Section 6. This vulnerability is primarily characterized by a decrease in lateral story stiffness and moment capacity in the longitudinal beams.

## 2. Typical Reinforced Concrete Frame School Buildings in Sri Lanka

The majority of school buildings in Sri Lanka consist of typical two- and three-story RC frame structures, as depicted in Figure 2. These buildings exhibit symmetry both in their floor plans and elevations. The floor plans for both types are rectangular, measuring 27.9 m in length and 9 m in width. Each floor plan includes eight classrooms, along with stairwells and storage rooms, as illustrated in Figure 3. The buildings feature a 9 m-wide single bay in the transverse direction and nine bays in the longitudinal direction, with each bay measuring 3.1 m in width. The height of a single story is also 3.1 m. Importantly, interior infill walls are present for partitioning, which can be oriented in the longitudinal direction, transverse direction, or both. Figures 4 and 5 provide reinforcement details for all beam and column sections of the two- and three-story school buildings, respectively. Notably, school buildings in Sri Lanka were primarily designed to withstand gravitational forces, and there is no apparent indication of a capacity-based seismic design approach being applied or considered.

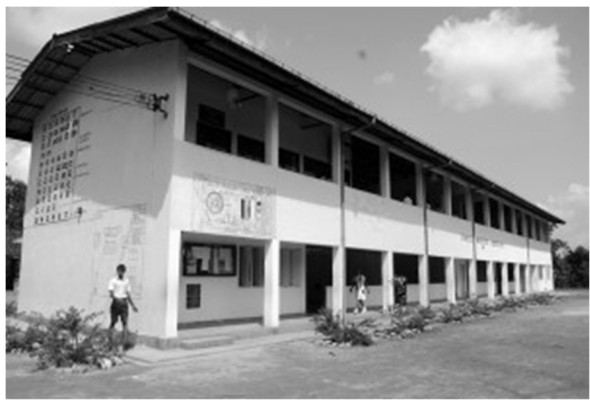 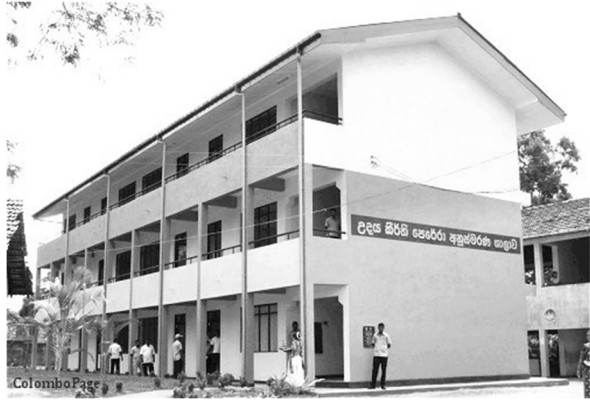

**Figure 2.** (**left**) Typical two-story school building and (**right**) a more classical three-story school building.

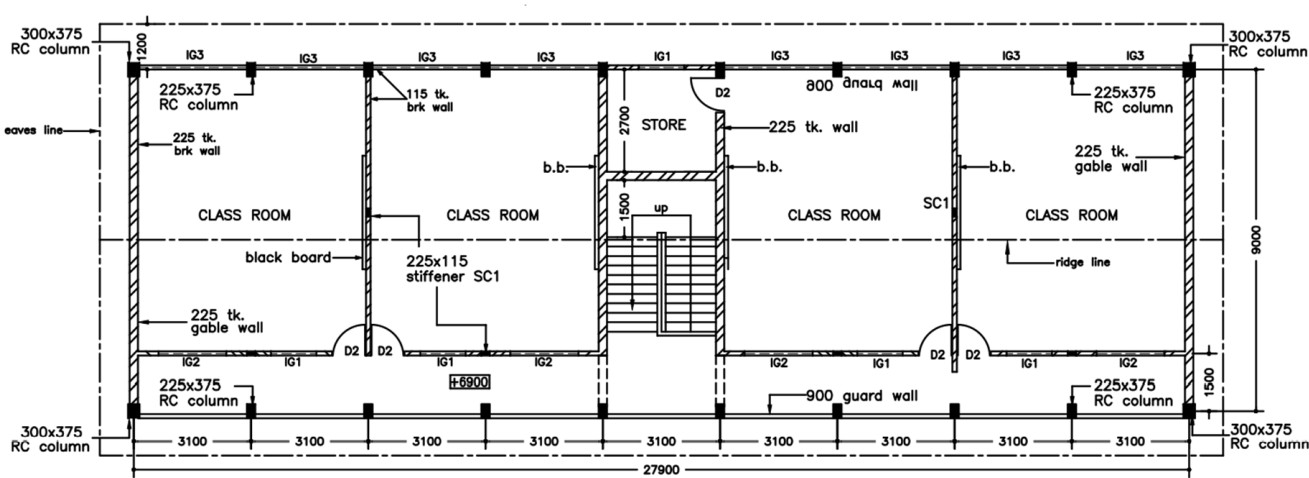

**Figure 3.** Typical plan view of the school buildings derived from an original drawing completed during design phase.

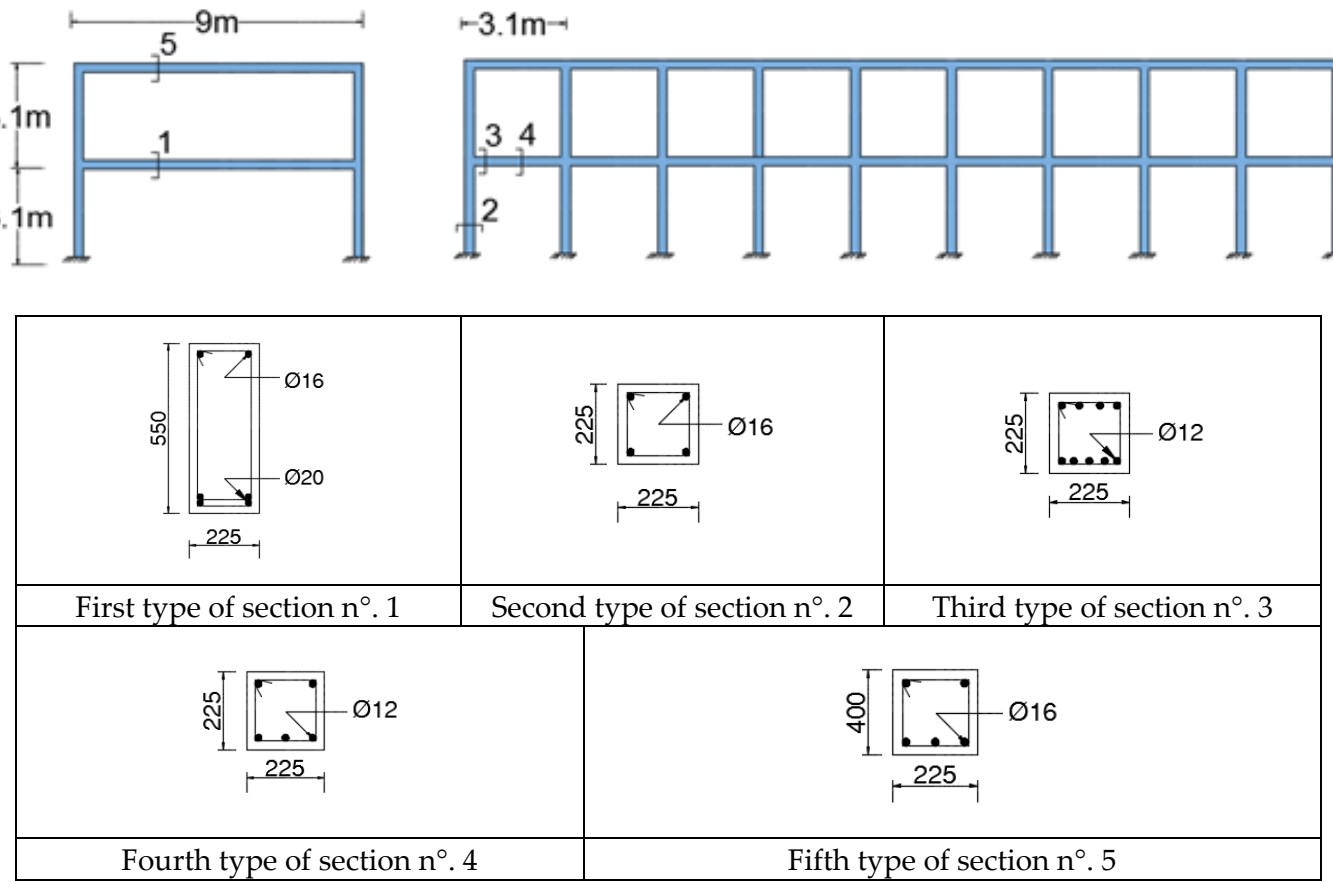

**Figure 4.** Reinforcement details of beams and columns in two-story school building.

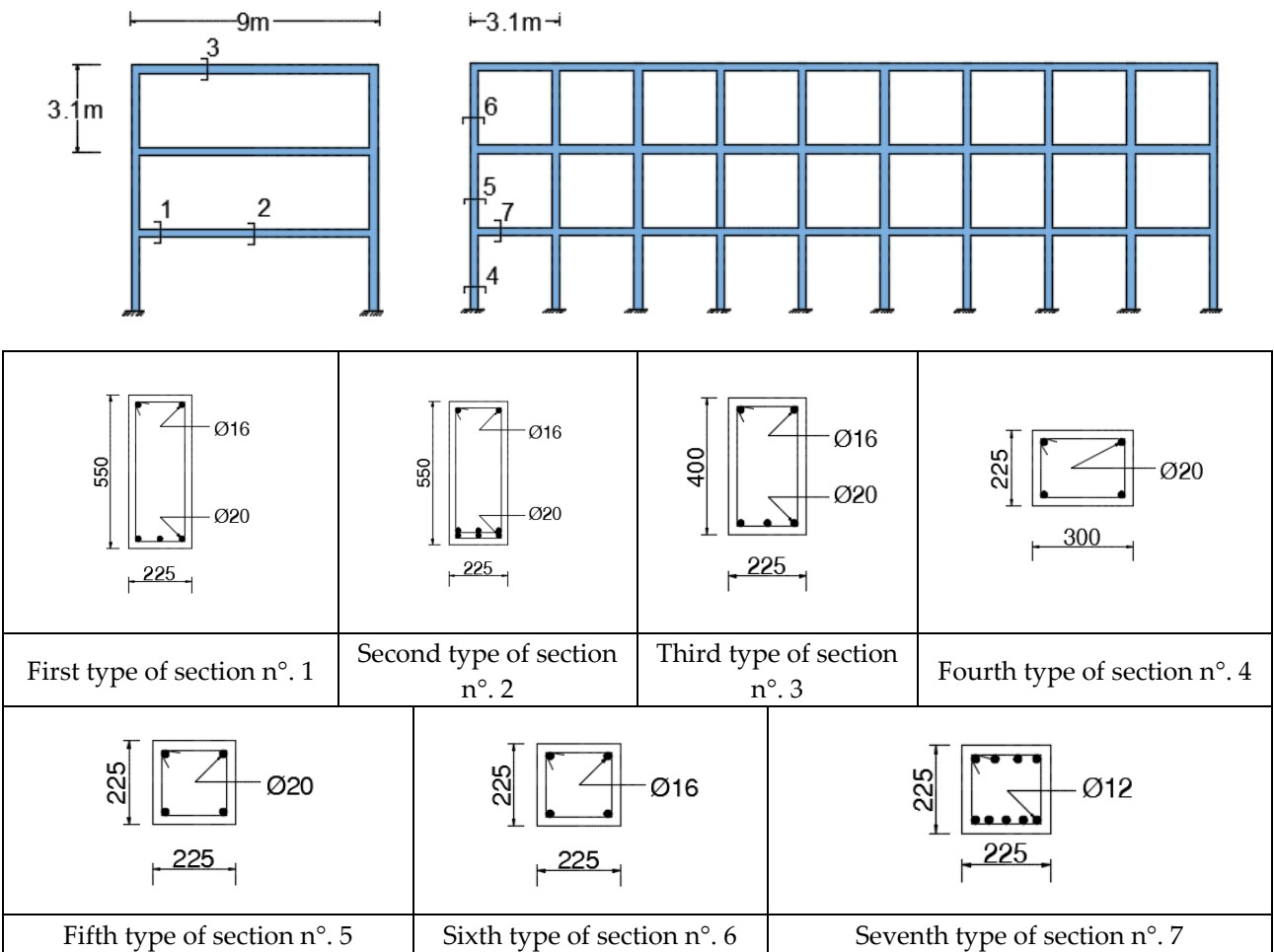

**Figure 5.** Reinforcement details of beams and columns in two three-story building.

## 3. Numerical Modelling of School Building

Advanced numerical modeling techniques encompass four primary types of analyses, as indicated in the main research reported by Marjanishvili [34]. These include linear static analysis, pushover (or nonlinear static analysis), linear response spectrum (or linear dynamic analysis), and nonlinear dynamic analysis (often referred to as time history analysis, which can be further refined into IDA). Linear static and dynamic analyses fall short of capturing critical aspects such as buckling, significant displacements/rotations, second-order effects (including p-delta effects), inelastic behavior, strength or stiffness degradation, and the development of concentrated plastic hinges (representing ductility and dissipation). Pushover analysis, while simpler, serves to generate a force-versus-displacement capacity curve akin to more advanced nonlinear dynamic analysis. This curve helps designers in assessing whether a structure possesses adequate capacity to withstand seismic forces or is prone to failure. Pretlove et al. [35] made a significant discovery, emphasizing that some buildings may be statically safe but dynamically unsafe. This discrepancy arises because time-dependent earthquake loads, resulting from the ultimate behavior of structural elements, can trigger the progressive failure of other elements before a new equilibrium state is achieved. This phenomenon, known as a cascade, pancake, or domino effect, underscores the importance of incorporating advanced nonlinear dynamic behavior and modeling into the numerical simulations of existing structures.

In light of the aforementioned findings, 3-D finite element models for the two school buildings were created using the OpenSees [36] software to conduct IDA. Figure 6 provides an illustration of these finite element models for both the two-story and three-story

buildings. In these models, the RC beams and columns are represented using inelastic beam–column elements [37], employing a force-based (FB) finite element formulation that offers several advantages over the conventional displacement-based (DB) formulation. One notable advantage is that this approach allows for the derivation of curvature distribution with a high degree of precision using only a single element along the entire member. This becomes even more numerically accurate when an appropriate number of integration points along the length of the member is selected. This methodology has been rigorously tested and proven to be numerically robust, efficient, and reliable, even in cases involving strength softening, such as observed in elements undergoing compression crushing [38]. Unlike linear analyses, where elastic properties can be assigned to a single element per member, nonlinear analyses need proper discretization of elements to accurately account for the distribution of inelastic behavior.

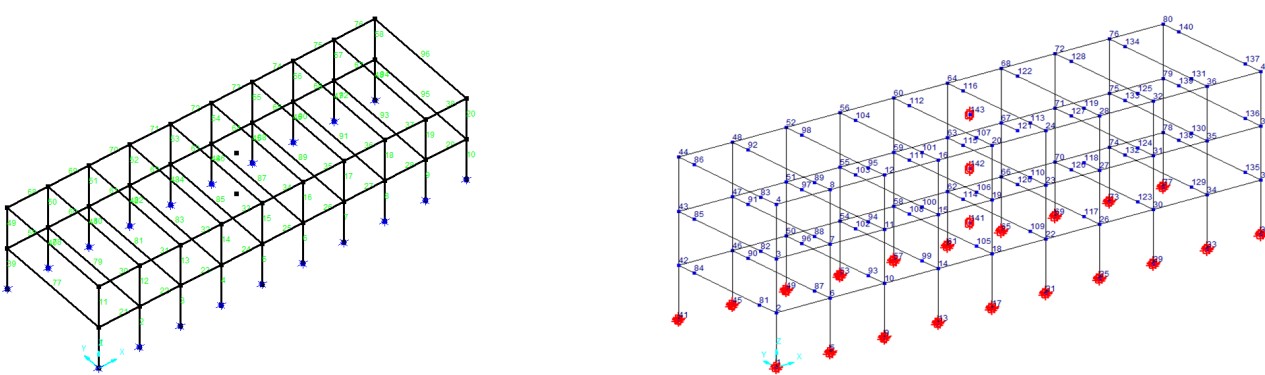

**Figure 6.** Numerical models of two- and three-story RC frame school buildings.

In general, a higher number of elements per member results in a more precise estimation of the global and local structural behavior. However, an excessive number of elements can significantly increase the computational workload required to perform the analyses. Furthermore, the adoption of FB elements enables the utilization of a single element per member, resulting in a substantial reduction in model size compared to using DB counterparts.

Past studies [39,40] have highlighted that the seismic performance of existing RC structures is influenced not only by the properties of the structural elements but also by certain non-structural elements that, while not directly involved in the lateral resistance system, can impact their behavior. Therefore, an essential consideration in modeling existing structures is how to represent the floor slabs of each level. In this regard, we modeled the floor slabs as rigid diaphragms, applying the rigid plane stiffness assumption. Employing rigid diaphragms for the floor slabs serves to simplify the model by reducing the number of degrees of freedom. We introduced a node at the geometric center of each floor, designating it as the master node, while all surrounding nodes were treated as slave nodes. To ensure stability, we applied a single-point constraint to fix the nodes at the base of the columns, restraining all six degrees of freedom. Additionally, this constraint was used to limit the rotational degrees of freedom of master nodes around the global x and y directions, as well as the vertical translation degree of freedom. The co-rotational approach is selected to describe moderate to large deformation effects of plastic rotation in inelastic beam–column elements, while the small deformation method is applied only for the evaluation of local stresses and strains of the inelastic beam–column elements. The concentration of mass at the beam–column joints represents a common approach that enables a significant reduction in the total number of model elements, leading to computationally more efficient analyses. The eigenvalue problem turns out to be more discrete, and it has also been attested [41] that the evaluation of mode shapes based on lumped masses is consistent with the choice of distributing the lateral pushover forces at the structural joints.

In this study, we employed FB fiber elements, with each element having five integration points. These integration points represent individual fiber sections, and their characteristics are determined by the cross-sectional area and steel reinforcement details of the element, as depicted in Figure 7. Each section of the element, whether it is a column or a beam, is divided into five quadrilateral patches, as shown in Figure 7. For the unconfined concrete region of the section, we divided it into ten by three divisions, creating a total of 120 fibers. In the case of the confined concrete region, we subdivided it into 10 by 10 divisions, resulting in 100 fibers. It is worth noting that each fiber was assigned the uniaxial stress–strain relationship of unconfined concrete.

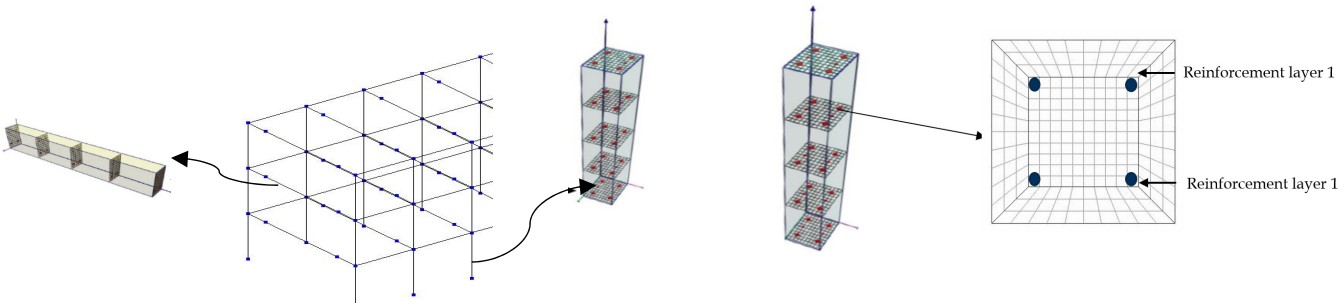

**Figure 7.** Integration points of elements and fiber discretization according to the FB approach.

The Kent–Park model was employed to represent the uniaxial nonlinearity of the concrete material [42]. This widely accepted concrete constitutive law incorporates a degraded linear unloading and reloading stiffness, following the influential research by Karsan and Jirsa [43]. It is important to note that tensile strength was disregarded in this model. Since there were no adequate steel reinforcements contributing to the confinement of the concrete core in the columns and beams of the two buildings, the confinement effect on the concrete section was greatly diminished (an 8 mm diameter stirrup every 300 mm along beams and columns). Equation (1) outlines the parabolic stress–strain relationship of concrete in compression up to the peak compressive stress, denoted as $f_c'$. The strain $\varepsilon_0$ corresponding to the peak stress is adopted equal to 0.002, which is a usually accepted assumption for unconfined concrete:

$$f_c = f_c' \left\{ \left[ \frac{2\varepsilon_c}{\varepsilon_0} \right] - \left[ \frac{\varepsilon_c}{\varepsilon_0} \right]^2 \right\} \tag{1}$$

where $\varepsilon_c$ is the evolving value of the strain that can reach the $\varepsilon_{cu}$ value, which is the ultimate strain level (the parabolic-plastic stress–strain curve for concrete used is based on the Portland Cement Association's parabolic stress–strain curve). The pronounced softening portion of the curve is supposed to be fully linear, and its slope is specified by evaluating the strain when concrete stress had fallen to 0.5 of the maximum stress (0.5 $f_c'$). Figure 8 illustrates the stress–strain relationship of confined and unconfined concrete in a first-story column. It is worth noting that tensile strength was not considered for both confined and unconfined concrete in this study.

The mass participation factor is 0.85 for the first sway mode of vibration in both directions.

The material nonlinearity of the reinforcement steel is represented by a uniaxial bilinear steel material model with kinematic hardening characterized by nonlinear evolution expressions. The yield strength, initial elastic tangent, and strain-hardening ratio are 460 MPa, 200 GPa, and 0.5%, respectively.

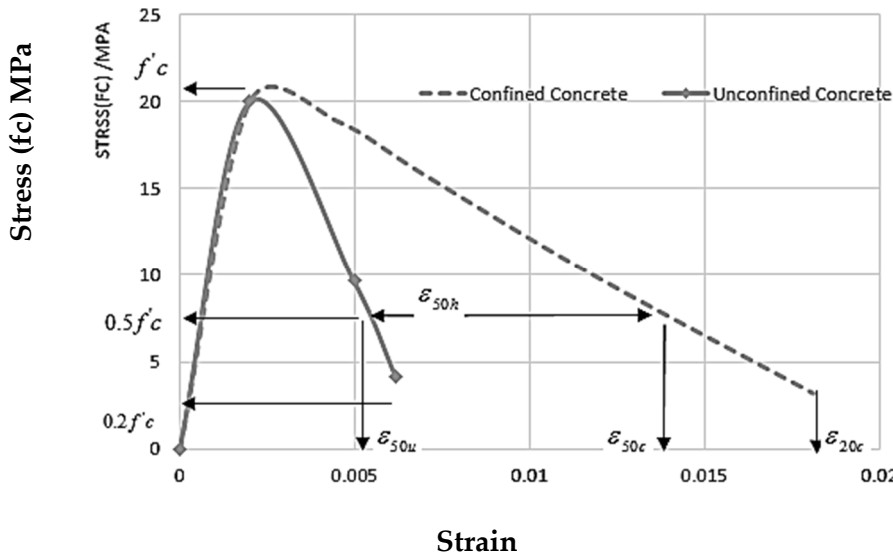

**Figure 8.** Stress–strain relationship of confined and unconfined concrete.

## 4. Selection of Ground Motion

To perform IDA for buildings using nonlinear dynamic analyses, one can choose either a set of artificially generated acceleration time histories (using stochastic algorithms) or natural acceleration time histories (selected from real earthquakes) [44,45]. However, in this study, we opted for thirty real ground motion records (Table 1) sourced from PEER databases, which were recorded in various locations worldwide [46]. Employing actual accelerograms as seismic input instead of synthetic ones offers a significant advantage as it incorporates the amplitude, frequency content, energy, and duration characteristics of genuine ground motion [46]. Each of these attributes plays a crucial role in the seismic risk assessment of buildings. Additionally, using ground motion data recorded during real earthquakes is preferred due to their inclusion of realistic low-frequency components and appropriate temporal correlation between horizontal and vertical motion components. Figure 10 displays the acceleration response spectra of the selected ground motion records. The steps involved in selecting accelerogram records based on seismic hazards in Sri Lanka are as follows:

1.  Consultation of seismic hazard studies and geological reports specific to Sri Lanka to gain insights into the seismic hazard in the region. Sri Lanka's proximity to the boundary between the Indian and Australian tectonic plates makes it susceptible to earthquakes. In particular, we referred to the study conducted by Venkatesan and Gamage [47], which provided hazard values in terms of peak ground acceleration and elastic spectral acceleration with a 5% damping ratio for return periods of 475, 975, and 2475 years, presented as raster maps. Additionally, we cross-referenced the results with the openquake application [48] and found a good match between the two sources;

2.  Establishing the specific seismic hazard levels required for our analysis, which, in this instance, were defined as return periods of 475 years;

3.  Determining the precise locations in Sri Lanka that were the focus of our seismic hazard assessment. According to the study in [47], the region around the capital city, Colombo, exhibits the highest anticipated PGA in rock sites, approximately 0.043 g for a 475-year return period;

4.  Assessing the target spectra (Figure 9) utilized in the process of selecting accelerogram records for seismic hazard analysis. These target spectra were chosen to accurately represent the anticipated ground motion levels for Colombo [49];

5. Identifying accelerogram records from earthquakes that align with the established criteria. Our goal was to find records from seismic events that effectively represent the seismic hazard present in Sri Lanka.

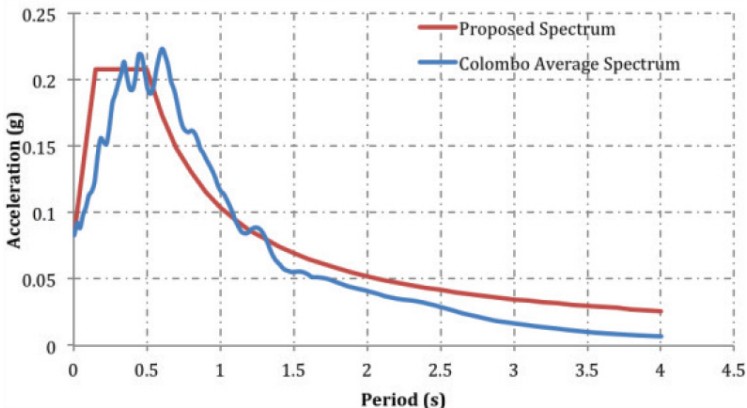

**Figure 9.** Bedrock response spectrum at Colombo.

**Table 1.** Selected ground motions and main properties.

| No | Event | Year | $M_w$ | $R_{jb}$ [km] | $R_{rup}$ [km] | $V_{s30}$ [m/s] | PGA [g] | PGV [cm/l] | PGD [cm] |
|----|-------|------|-------|---------------|----------------|-----------------|---------|------------|----------|
| 1 | Whittier Narrows-01 | 1987 | 5.99 | 14.9 | 20.8 | 245 | 0.3 | 37.6 | 4.9 |
| 2 | Irpinia, Italy-01 | 1980 | 6.9 | 22.5 | 22.6 | 561 | 0.22 | 14.2 | 3.2 |
| 3 | Whittier Narrows-01 | 1987 | 5.99 | 4.5 | 17.4 | 368 | 0.3 | 20.9 | 3.1 |
| 4 | Whittier Narrows-01 | 1987 | 5.99 | 18.3 | 23.4 | 267 | 0.33 | 27 | 5 |
| 5 | Loma Prieta | 1989 | 6.93 | 24.3 | 24.6 | 240 | 0.17 | 25.9 | 12.6 |
| 6 | Friuli, Italy-01 | 1976 | 6.5 | 15.0 | 15.8 | 505 | 0.35 | 22 | 4.1 |
| 7 | Kocaeli, Turkey | 1999 | 7.51 | 68.1 | 69.6 | 175 | 0.25 | 40 | 28.4 |
| 8 | Loma Prieta | 1989 | 6.93 | 8.8 | 9.6 | 1428 | 0.41 | 31.6 | 6.3 |
| 9 | Loma Prieta | 1989 | 6.93 | 13.8 | 14.3 | 222 | 0.42 | 38.7 | 7.1 |
| 10 | Loma Prieta | 1989 | 6.93 | 17.9 | 18.3 | 663 | 0.13 | 12.7 | 4.7 |
| 11 | Loma Prieta | 1989 | 6.93 | 79.7 | 79.8 | 584 | 0.23 | 38 | 11.4 |
| 12 | Coalinga-01 | 1983 | 6.36 | 23.8 | 24.0 | 275 | 0.23 | 23.6 | 5.8 |
| 13 | Imperial Valley-06 | 1979 | 6.53 | 22.0 | 22.0 | 242 | 0.35 | 33 | 19 |
| 14 | Kobe, Japan | 1995 | 6.9 | 22.5 | 22.5 | 312 | 0.34 | 27.7 | 9.6 |
| 15 | San Fernando | 1971 | 6.61 | 22.2 | 27.4 | 425 | 0.19 | 5.6 | 0.9 |
| 16 | San Fernando | 1971 | 6.61 | 14.0 | 19.3 | 602 | 0.37 | 16.8 | 1.6 |
| 17 | Northridge-01 | 1994 | 6.69 | 20.1 | 20.7 | 450 | 0.57 | 51.9 | 9 |
| 18 | Chalfant Valley-01 | 1986 | 5.77 | 23.4 | 23.5 | 303 | 0.13 | 8.5 | 2.4 |
| 19 | Tabas, Iran | 1978 | 7.35 | 0.0 | 13.9 | 472 | 0.333 | 20.4 | 11.6 |
| 20 | Duzce, Turkey | 1999 | 7.14 | 9.1 | 9.1 | 338 | 0.11 | 11.2 | 9.8 |
| 21 | Nahanni, Canada | 1985 | 6.76 | 0.0 | 4.9 | 605 | 0.32 | 33 | 6.6 |
| 22 | Parkfield | 1966 | 6.19 | 9.6 | 9.6 | 290 | 0.37 | 21.8 | 3.8 |
| 23 | Corinth, Greece | 1981 | 6.6 | 10.3 | 10.3 | 361 | 0.24 | 23.4 | 11.6 |
| 24 | Spitak, Armenia | 1988 | 6.77 | 24.0 | 24.0 | 344 | 0.2 | 28.6 | 9.7 |
| 25 | Santa Barbara | 1978 | 5.92 | 0.0 | 12.2 | 515 | 0.2 | 16.3 | 3 |
| 26 | Landers | 1992 | 7.28 | 23.6 | 23.6 | 354 | 0.24 | 51.4 | 43.8 |
| 27 | Kozani, Greece-01 | 1995 | 6.4 | 14.1 | 19.5 | 650 | 0.22 | 9.3 | 1.7 |
| 28 | Northern Calif-01 | 1941 | 6.4 | 44.5 | 44.7 | 219 | 0.12 | 6.3 | 1.1 |
| 29 | Superstition Hills-02 | 1987 | 6.54 | 13.0 | 13.0 | 194 | 0.17 | 23.4 | 13 |
| 30 | El Alamo | 1956 | 6.8 | 121.0 | 121.7 | 213 | 0.05 | 6.6 | 5 |

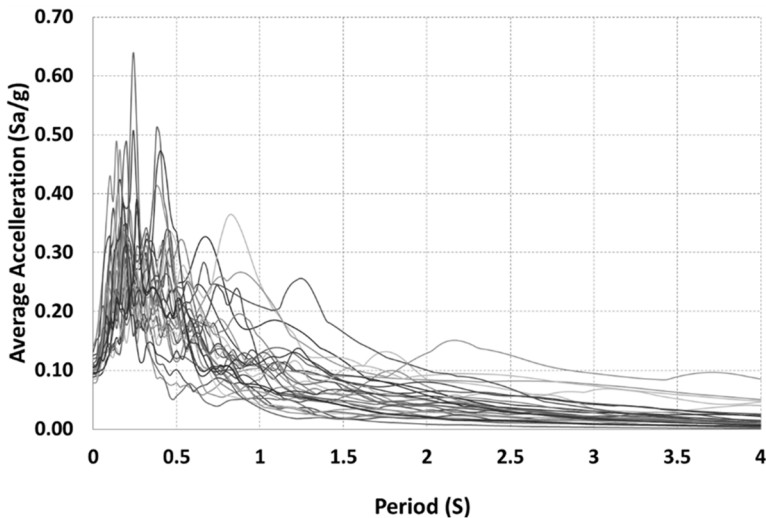

**Figure 10.** Acceleration response spectrum of selected ground motions.

## 5. Incremental Dynamic Analysis and Performance Levels

In this study, the primary aim of conducting IDA is twofold: first, to formulate demand models, and second, to determine inter-story drift limits for various performance objectives within two distinct buildings. This process involves scaling up each chosen ground motion record, transitioning from low to high intensities, as elaborated in Vamvatsikos and Cornell [31–33]. The structural response is quantified by assessing the maximum inter-story drift ratio, while the ground motion characteristics are evaluated by analyzing the spectral acceleration at the first mode period. The IDA curves were established using analysis data, employing PGA in $g$ as the intensity measure (IM) and drift of the resisting frames as the damage measure. Figure 11 illustrates the nonlinear IDA curves for the two case studies, specifically for the two- and three-story buildings. These curves are generated based on real ground motions (Table 1) that have been carefully selected, with the maximum allowable drift serving as the critical limit state.

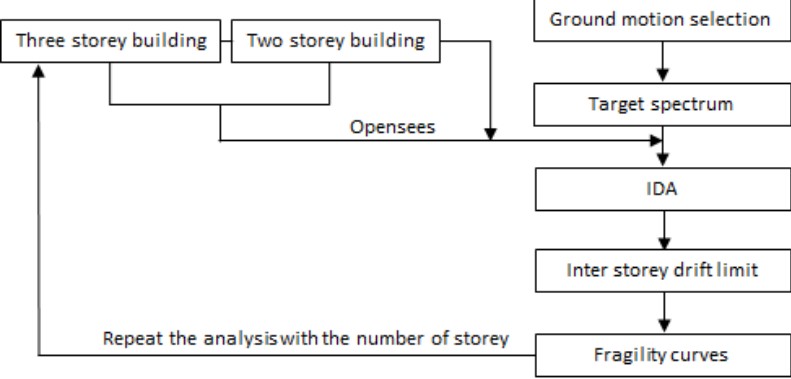

**Figure 11.** Flowchart for fragility derivation: IDA procedure.

The maximum inter-story drift corresponding to a particular intensity level is determined by applying the accelerogram in the longitudinal direction of the model. This typically results in a higher value compared to the drift observed in the transverse direction. This disparity arises because the infill brick walls effectively act as bracing elements in the transverse direction. Figures 12 and 13 display the IDA curves for 30 carefully selected ground motion records. These curves pertain to a two-story building and a three-story building, respectively. As the IDA curves progress towards the right, signifying an increase in the PGA, the maximum drift of the two frames also increases until reaching the defined limit state, as detailed in Table 2.

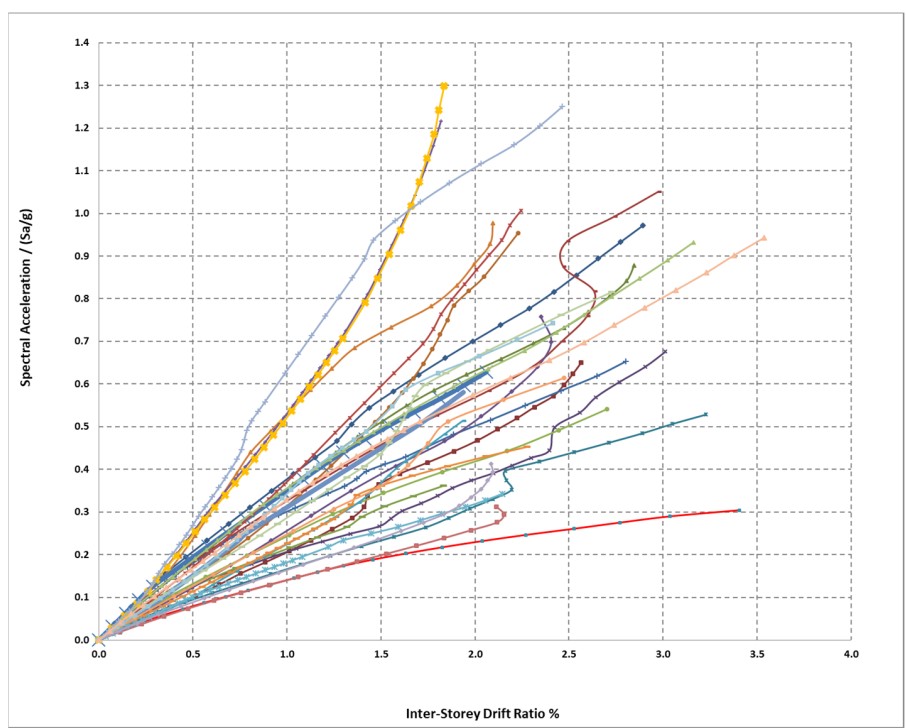

**Figure 12.** IDA curves for 30 earthquakes for two-story building.

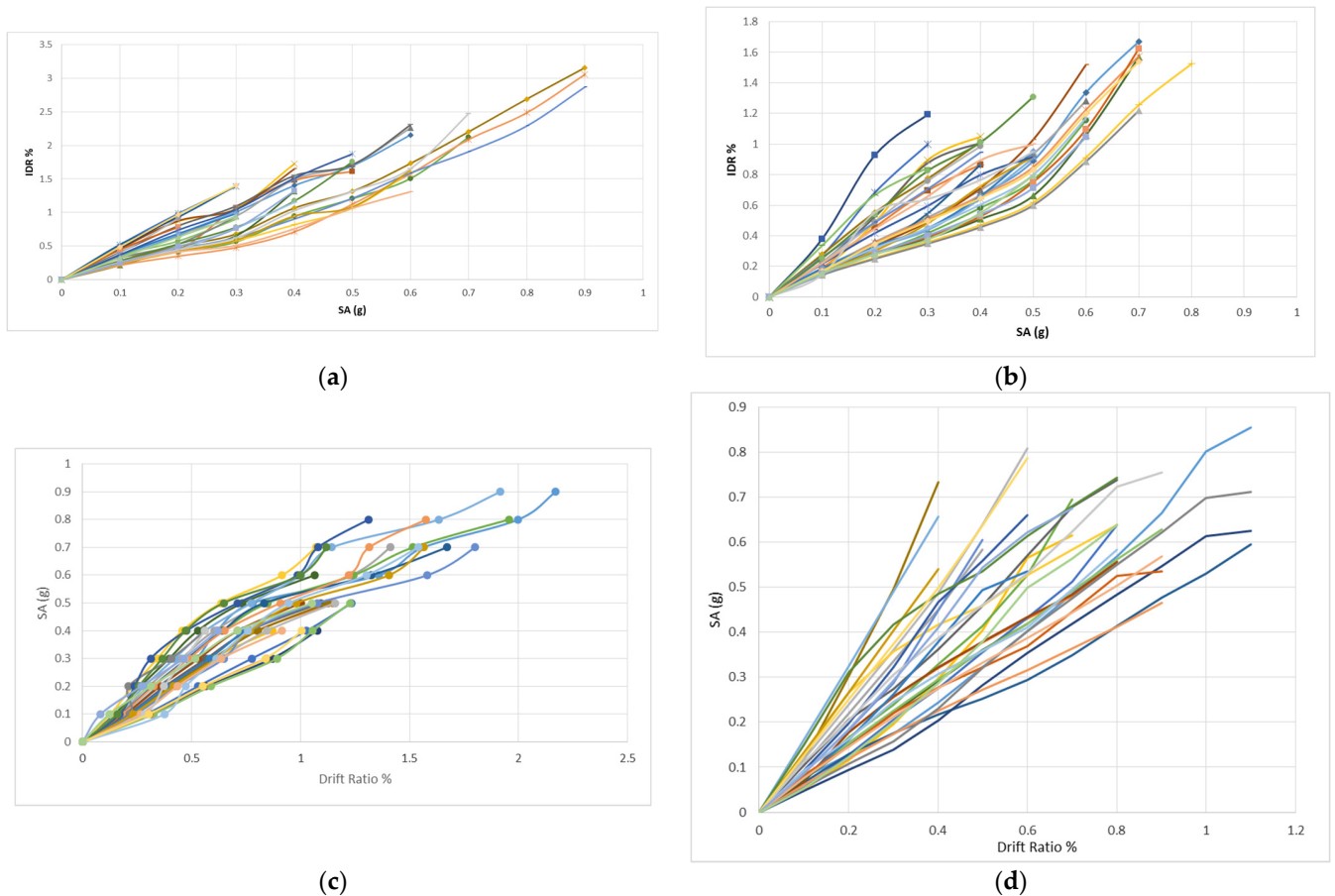

**Figure 13.** IDA curves for 30 earthquakes for three-story buildings: (**a**,**b**) bare frames in x and y direction respectively and (**c**,**d**) frames with infill panels always in x and y direction respectively.

**Table 2.** Inter-story drift ratios for different performance levels.

|  | IO Performance Level | CP Performance Level |
|---|---|---|
| Inter-story drift ratio (two storys) | 0.80 | 1.75 |
| Inter-story drift ratio (three storys) | 1.03 | 3.43 |

In this study, we have considered two distinct performance levels: IO and CP [50]. The IO performance level is defined as the structural state at its elastic limit. Conversely, the CP performance level is established based on the specific failure mode observed in critical elements, where larger plastic deformations are anticipated. For our building, we have identified the IO performance point on each IDA curve, which corresponds to the flexural yielding at the first-story beam. IO is a vital performance objective, particularly for structures like schools, when facing the threat of an earthquake. Essentially, it means that after the seismic event, the building should remain safe for occupancy without incurring significant structural damage or posing any danger to its occupants. In practical terms, this ensures that people should be able to stay inside the building and continue their normal activities immediately after the earthquake has subsided. This objective is typically applied to critical structures, such as schools, located in regions with low to moderate seismic risk, such as Sri Lanka.

Drawing upon the research conducted by Vamvatsikos and Cornell [31], the CP performance point on an IDA curve is identified as the point where the IDA curve demonstrates a slope equivalent to 20% of the elastic stiffness while also residing within a softening branch of the curve. However, this particular study takes a slightly different approach by incorporating element performance to define the CP performance point on the IDA curve. From the results of the nonlinear dynamic analyses conducted to establish the IDA curves depicted in Figures 12 and 13, we can deduce the following findings and implications:

- The ultimate structural failure of both buildings can be attributed to the failure of the first-story beam elements in flexure, primarily due to excessive deformation;
- Because the column spacing is considerably narrower in the longitudinal direction compared to the transverse direction, the resultant beam sections of the longitudinal beams, derived from the gravity design of the frames, exhibit significantly lower strength and stiffness in comparison to the corresponding column sections. In fact, these longitudinal beams are the weakest components within the moment-resisting frames in the longitudinal direction, leading to a concentration of plastic deformations primarily at the first-story beams, as illustrated in Figure 14;
- The points of global failure on the IDA curves for the buildings are associated with a 20% reduction in moment capacity for all first-story beam elements. This reduction is established by utilizing moment-curvature curves derived from the structural analysis. Likewise, the calculated drift ratio values for each earthquake are averaged to provide a normalized result and are presented in Table 2;
- It is noteworthy that the two-story frame exhibits superior resilience and can withstand higher PGA in comparison to the three-story frame;
- These findings underscore the necessity for structural retrofitting in structures of this kind to mitigate the risk of structural collapse during seismic events.

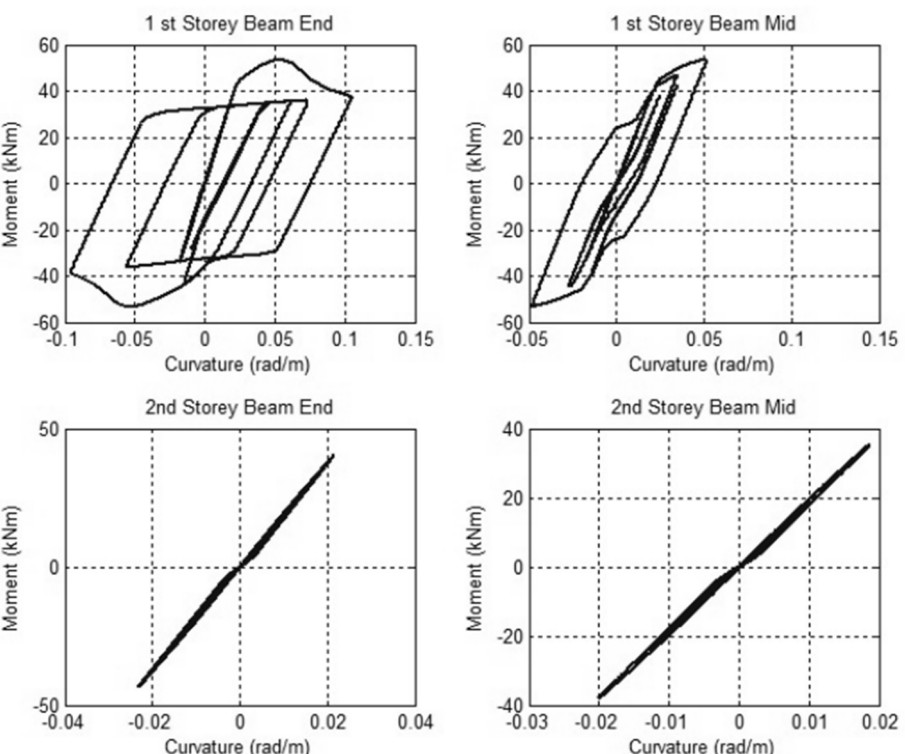

**Figure 14.** Moment curvature diagrams of two-story building.

## 6. Exploring Probabilistic Seismic Demand and Fragility Analysis

A Probabilistic Seismic Demand Model (PSDM) serves as a mathematical framework that characterizes the likelihood distribution of structural demands contingent upon a specific IM. These structural demands are commonly measured using chosen metrics like inter-story drift or ductility. In their referenced work [51], Cornell and his colleagues introduced the concept that the median demand estimate ($\widehat{D}$) can be expressed through a power model as follows:

$$\widehat{D} = a \cdot IM^b \tag{2}$$

In this context, *IM* represents the chosen seismic intensity measure, while *a* and *b* denote the regression coefficients. For this study, *IM* is defined as the spectral acceleration at the first mode period of the structure, and the selected demand parameter is the inter-story drift ($\theta_{max}$).

The study's demand model for each school building is formulated based on the IDA curves derived from 30 ground motion records. These curves plot the median inter-story drift values against spectral acceleration at intervals of 0.1 g for the first mode period. Subsequently, through a power regression analysis of inter-story drift against spectral acceleration, as illustrated in Figures 15 and 16, the PSDM parameters *a* and *b* are determined for two-story and three-story school buildings, respectively. Table 3 provides a summary of the coefficients *a* and *b* acquired for these two buildings.

**Table 3.** Regression coefficients of the power model expressed in Equation (2).

|  | *a* | *b* |
|---|---|---|
| Two-story building | 3.0776 | 1.2545 |
| Three-story building | 3.1545 | 1.0060 |

Additionally, it is a common practice to make the assumption that the distribution of the demand around its median adheres to a two-parameter lognormal probability distribution. Consequently, the variability ($\beta_{D|IM}$) of the demand in relation to its median

can be computed and is contingent upon the chosen intensity measure *IM* in Equation (2). It is noteworthy that this variability is regarded as constant within the range of *IM* values under consideration.

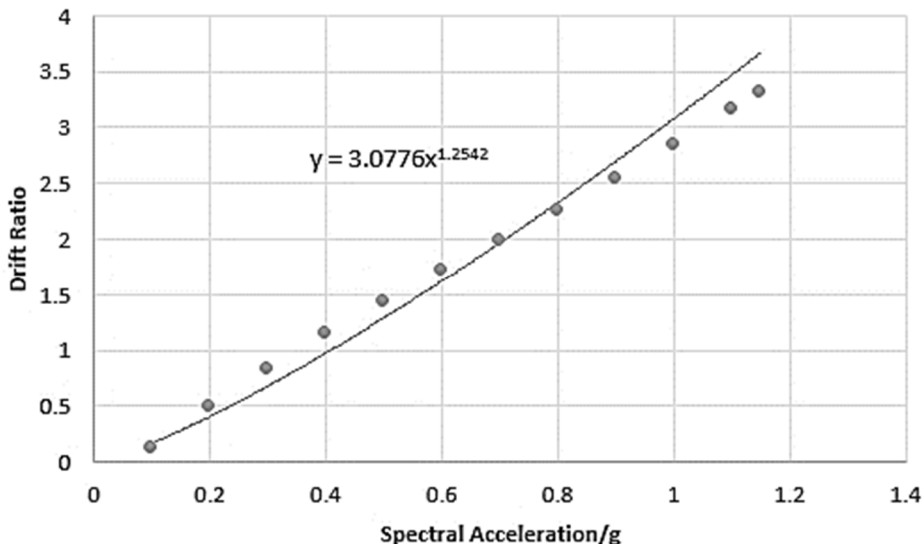

**Figure 15.** PSDM for two-story building.

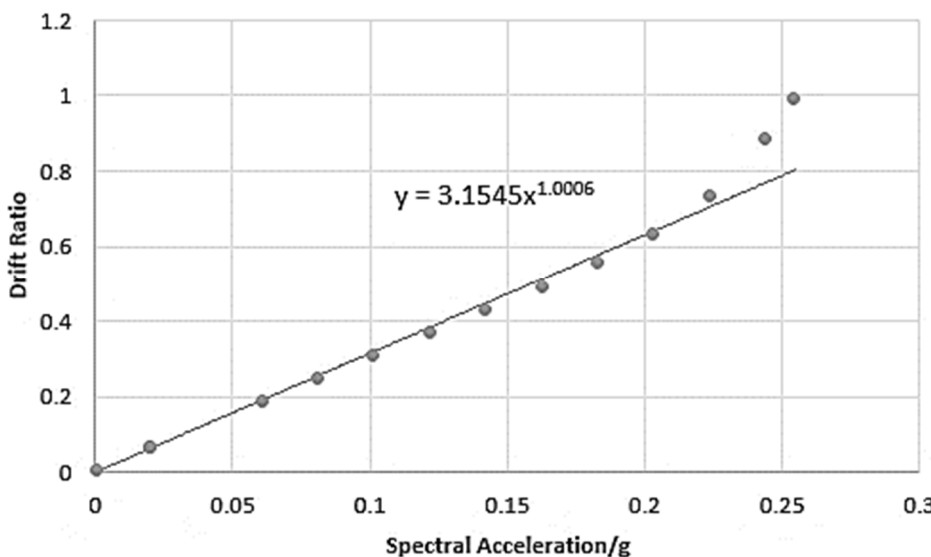

**Figure 16.** PSDM for three-story building.

Fragility can be succinctly defined as the probability that the seismic demand (*D*) imposed on a structure exceeds the capacity (*C*) of the structure. This probability statement is contingent upon a selected *IM*, which signifies the magnitude of the seismic loading. The general expression for this conditional probability is typically represented as

$$P(D > C|IM) \tag{3}$$

An approach to evaluate the fragility function, as outlined in Equation (3), involves convolving the PSDM with a capacity distribution. As mentioned earlier, assuming a

lognormal distribution for the demand at each level of the *IM*, the conditional probability can be expressed as follows:

$$P(D > C|IM) = 1 - \Phi\left(\frac{\ln(\hat{C}) - \ln\left(a \cdot IM^b\right)}{\beta_{D|IM}}\right) \tag{4}$$

Here, $\hat{C}$ represents the median structural capacity corresponding to the specified limit state. In this research, the seismic *IM* employed is the spectral acceleration at the fundamental period of the frame with 5% damping. Figures 17 and 18 provide visual representations of the fragility curves for the two-story and three-story school buildings chosen in this study, focusing on IO and life CP objectives.

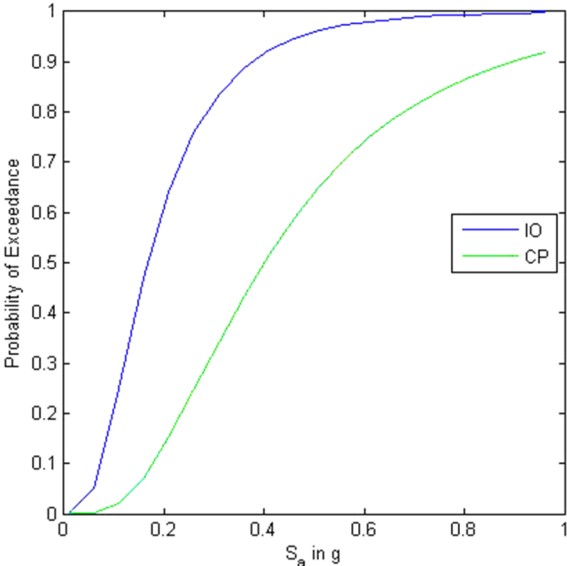

**Figure 17.** Fragility curve for two-story building at the IO and CP objectives.

Several observations can be made regarding the obtained fragility functions for two- and three-story school buildings, in alignment with the previously discussed hazard in Sri Lanka:

- The region around Colombo emerges as the most vulnerable area in the country [47] concerning seismic activities. Colombo is anticipated to experience a maximum expected PGA exceeding 0.043 g in a 475-year return period. In contrast, the rest of the country is exposed to relatively minor ground motions that disperse uniformly across the region, leading to its classification under the low seismicity category. The results presented in Figures 17 and 18 are specifically derived for the most vulnerable location, utilizing the most vulnerable and strategically significant buildings within that area;

- When comparing the pushover curves depicted in Figures 17 and 18 with the established trends for newly designed buildings [50], a noteworthy observation emerges. It becomes evident that the seismic vulnerability of low-ductile RC frames, which were not originally designed to withstand earthquake loads, remains a consistent concern for countries characterized by low-to-medium seismicity, such as Sri Lanka. This finding may also be extended to similar nations like Malaysia and Australia, particularly in regions like Tasmania. It is worth noting that these types of structures constitute the majority of existing school buildings in Sri Lanka. Consequently, evaluating their safety levels becomes a critical consideration for authorities when planning retrofitting measures;

- The distribution of results within the fragility curves suggests that the number of stories only has a marginal impact on damage probabilities. Despite the distinct and

unique structural characteristics of each building, there is a general trend of increasing damage probabilities as the number of stories increases. Additionally, it is noteworthy that the disparities between the IO and higher damage cases (CP) are considerably pronounced in both two- and three-story school buildings;

- The observed variation in fragility from two to three stories indicates that the number of stories is not as influential as other parameters for low to mid-rise buildings. In essence, the most influential factors that distinctly characterize or differentiate the fragility curves are the age of construction and outdated design processes. It is important to note that this conclusion holds true for all considered damage cases, including IO and CP;

- It is apparent that the bare frame buildings fully infilled are more vulnerable compared to bare frame typology.

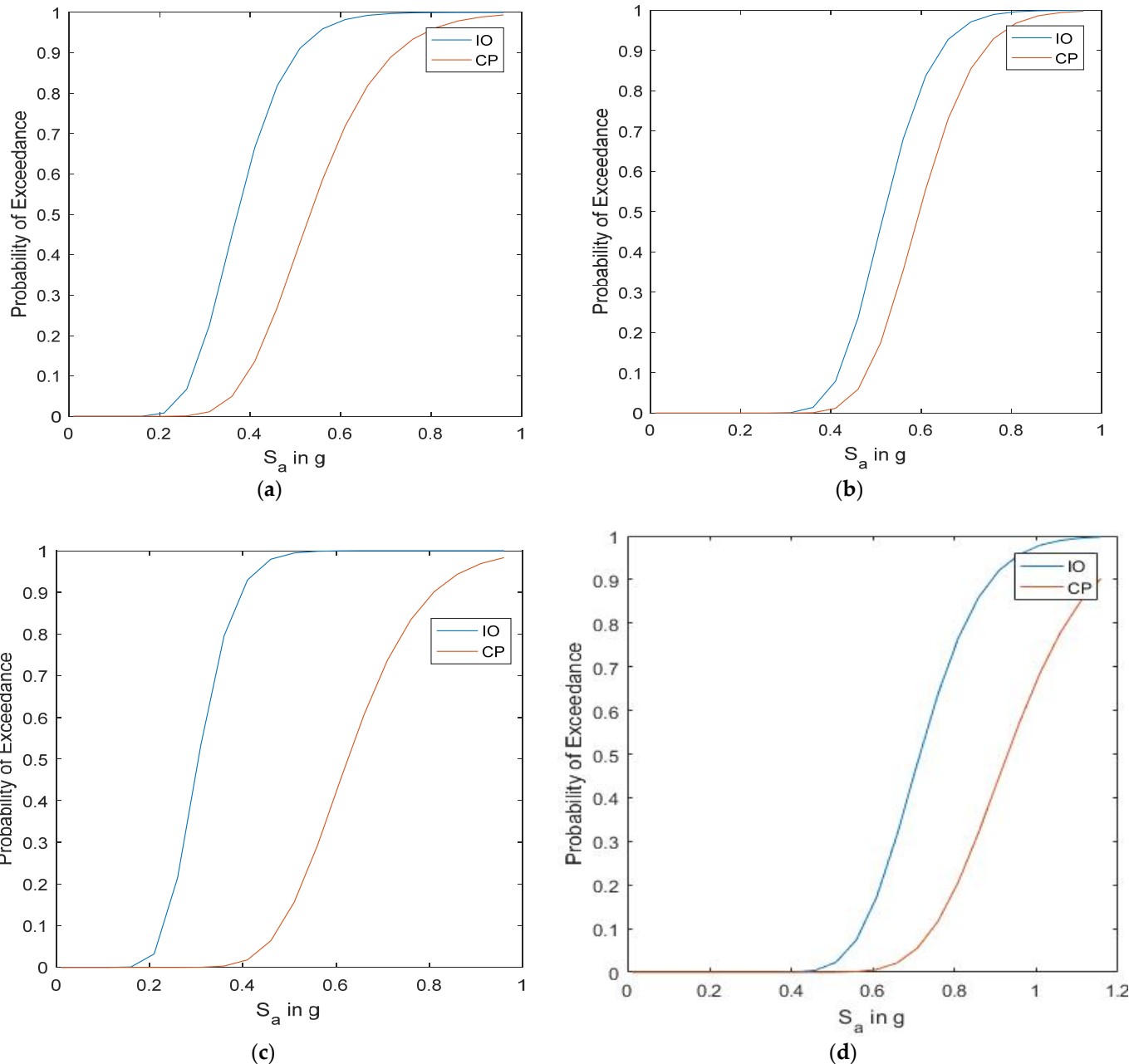

**Figure 18.** Fragility curve for three-story building at the IO and CP objectives: (**a**,**b**) bare frames in x and y direction respectively and (**c**,**d**) frames with infill panels always in x and y direction respectively.

## 7. Conclusions

In this research, we evaluated the seismic vulnerability of low ductile RC schools with a frame typology. We employed the IDA approach and fragility curve analysis. We specifically chose two different structural models, one with a two-story and the other with a three-story RC frame. These structures, really existing and inspected, were designed according to prevalent practices in Sri Lanka, adhering to outdated static codes. We utilized both IDA and pushover analysis techniques to ascertain the drift capacities of these selected structures. A total of 30 earthquake records were meticulously chosen for this study. We focused on two distinct structural performance levels, namely IO and CP, for evaluating the structural elements. The following key findings emerged from our analyses and the fragility curves generated:

1. Analysis of the IDA curves reveals a clear vulnerability in the longitudinal direction for both the two- and three-story RC school buildings. This vulnerability is marked by a notable reduction in lateral story stiffness and a diminished moment capacity in the longitudinal beams;

2. Furthermore, the significant rotation observed in the beam–column joints at the first-story level ultimately leads to the failure of the longitudinal beams. This failure occurs as plastic hinges form at the junction of the beam and beam–column joint;

3. Upon comparing the pushover curves with the well-documented trends in the scientific literature concerning newly designed buildings, a significant observation comes to light. It becomes increasingly apparent that the seismic vulnerability of low-ductile RC frames, initially not intended to withstand earthquake forces, continues to be a persistent concern in regions with low-to-medium seismic activity, such as Sri Lanka;

4. In accordance with the design spectrum specific to Sri Lanka, the spectral acceleration values for the first mode periods hover around 0.25 g. Under this level of excitation, the likelihood of achieving the IO and CP performance objectives for two-story school buildings is approximately 75% and 25%, respectively. Meanwhile, for three-story school buildings, these probabilities are approximately 50% and 10%, respectively. Hence, it can be deduced that introducing a section with an increased moment capacity for the longitudinal beams (while still maintaining a moment capacity lower than that of the column in the longitudinal direction) can effectively delay structural failure. This modification would enhance the building's capacity to withstand more substantial earthquakes and significantly improve its overall seismic performance;

5. Despite the individual and distinctive structural attributes of each building, there exists a broad trend of heightened damage probabilities as the number of stories increases. Moreover, it is important to highlight that the disparities between the IO and more severe damage cases such as CP are quite pronounced in both two- and three-story school buildings;

6. Of paramount importance, the damage index that approaches the CP performance level is evident in the case of the two-story school building when subjected to an earthquake with a PGA of 0.52 g. Likewise, an IO performance level is attained when the earthquake registers a PGA of 0.33 g. As for the three-story school buildings, the damage index approaches the CP performance level during an earthquake with a PGA of 1.1 g, and an IO performance level is reached at a PGA of 0.35 g;

7. A limitation of this research is the relatively small number of buildings included in the study. This limited sample size may not comprehensively represent the diversity of building types and construction techniques found in larger urban contexts. Consequently, this study's findings and conclusions open the door to more advanced and refined analyses and provide valuable insights into the future direction for assessing these buildings;

8. Another limitation of this study is the lack of comprehensive in situ inspections and on-site experimental testing of the selected buildings. This absence impedes the acquisition of a more in-depth understanding of the mechanical properties of the construction materials used in these structures. Consequently, accurately assessing

the structural performance and vulnerabilities of the buildings in the study is challenging. Nevertheless, it is worth noting that the mechanical properties employed were based on well-established industry practices during the time of construction. This insight paves the way for potential future developments utilizing non-destructive inspection techniques, which can further enhance our understanding and assessment of these structures.

These findings emphasize the crucial significance of engineering school buildings to effectively withstand the lateral forces generated by seismic events. Additionally, they serve as essential inputs for shaping seismic rehabilitation plans and estimating the necessary budget to enhance the safety levels of existing low-ductile RC frames. The next phase of this study could encompass strengthening structural components through various means such as concrete jacketing, steel jacketing, Fiber Reinforced Polymer (FRP) wrapping, the application of pre-stress components, and the implementation of dampers.

**Author Contributions:** Conceptualization, T.M.A. and K.K.W.; methodology, K.K.W.; software validation, R.N.; formal analysis, T.M.A.; data curation, R.N.; writing—original draft preparation, T.M.A. and K.K.W.; writing—review and editing, R.N. All authors have read and agreed to the published version of the manuscript.

**Funding:** This research received no external funding.

**Institutional Review Board Statement:** Not applicable.

**Informed Consent Statement:** Not applicable.

**Data Availability Statement:** No data available.

**Conflicts of Interest:** The authors declare no conflict of interest.

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
