# Peer review of "Seismic Risk Assessment of Typical Reinforced Concrete Frame School Buildings in Sri Lanka"

_buildings, doi:10.3390/buildings13102662_

Round 1
Reviewer 1 Report
The manuscript investigates the seismic risk for two typical school buildings in Sri Lanka through the use of IDA analysis. The above-mentioned schools are designed only for gravitational forces. The paper concludes about the seismic risk and fragility curves are presented, as well, for such structures. The following observations are provided:
[1] The subject of the paper is a typical one. It is clear that framed structures designed without taking into account the concept of capacity design is vulnerable. The probabilistic results for such schools are non sense, because from the scratch it is well known that such structures are not resilient to medium and strong seismic actions.
[2] It is well known that such structures strongly interact with the infill masonry. In many cases of strong earthquakes they survive due to the seismic energy absorption from the infill masonry. Moreover, in case of low earthquakes the infill masonry provides to the flexible frame with stiffness. Therefore, it is important to have a comparative view through an analysis which takes into account also the influence of the infill masonry related to the seismic risk assessment.
Summing up, the paper will be revised by performing a series of IDA analysis considering the interaction between the infill masonry and the frame. Towards this direction, the Authors should compare the results of a bare frame and with one of the contribution of the infill masonry. Only in this way we can predict in stable manner the seismic risk of the buildings under consideration.
Author Response
We wish to thank the reviewer. All the answers to reviewer's comments are in the attached file. Please see the attachment.

Reviewer 2 Report
Please see attached file 'buildings-2640770 Comments01.pdf'.

Author Response

(The authors gave the same response as above.)

Reviewer 3 Report
The manuscript titled as “Seismic Risk Assessment of Typical Reinforced Concrete Frame School Buildings in Sri Lanka” derives fragility curves using the nonlinear dynamic analysis for the school buildings in Sri Lanka. Paper is superficial in terms of discussing the results and there are open issues to be highlighted in the manuscript. Paper is not suitable in its current form and it has to be improved considerably. Major concerns stated below:
1. Introduction section is should be re-evaluated and written. There are many statements not supported with references. In addition, it is like each paragraph is separate and have no coherence. Authors should clearly describe what they have done with supporting references and what this study fills the gap in the literature. Some important conclusions can be also mentioned at the end of this section.
2. Section 3.2 is too long and should be shortened. It not necessary to give long sentences or equation for the well-known concrete behavior. Accordingly, Section 3.1. and 3.2 can be combined.
3. Section 4 is superficial. Authors claimed that Sri Lanka has new probabilistic seismic hazard map which in turn means new target spectra for the region. However, record selection based on seismic hazard is not mentioned in the manuscript. Probabilistic assessment for these buildings is crucial relating with seismic hazard.
4. Determination of different performance states is important issue and it can considerably effect the fragility curves and performance assessment of buildings which is main interest of this study. Determination of performance level is not adequately discussed. Possible issues and consequences on the results should be clarified in the manuscript.
5. Paper in its current form gives the modeling issues and ends with describing the determination of fragility curves for different performance states. Authors should discuss about the obtained results considering the seismic hazard of Sri Lanka. These buildings may be in different regions of the country which has different seismicity. So, what is the meaning or what readers should infer from the exceeding probabilities of these limit states for different regions of the country. Authors can also compare the damage probabilities of two- and three- story school buildings. Some discussions are needed.
6. Transverse reinforcement details of the members are not given in the manuscript. Authors should give brief information or plot in Fig. 4.
7. Related with previous comment, authors have stated that they have used force-based element in Opensees, this model includes the shear deformation of members? Authors claimed that members were not suffered from shear, but bending as collapse prevention is described by beam bending deformations. Needs clarification!
8. Soma properties of buildings such as mass, modal participation factor buildings etc. should be provided for the readers.
9. Conclusion section should be revised also considering uncertainties associated with modeling, record selection, code-based evaluations for further studies to be done accounting Sri Lanka code improvements.
10. Additional comments and suggestions are provided in the attached file for authors.

The use of language is good in general and minor editing of English language required. Some annotations are provided for authors in attached manuscript.
Author Response

(The authors gave the same response as above.)

Reviewer 4 Report
Dear Authors,
Thank you for your attempts for presenting this work. There still exist some serious issues as follow:
• The authors are recommended to seek the services of a professional copywriter or native speaker for thorough corrections. Some sentences are so long and it is difficult to understand.
• The results of research should be quantified in the abstract.
• The history of research should be expressed chronologically.
• The flowchart should be added.
• The history should be added until 2023.
•The theoretical fundamentals should be stated completely.
•The authors should clearly explain this research's originality aspects that make this paper valuable for publication. This research doesn't have a novelty.
• When you used abbreviations, you should not use the complete format. For example: Incremental dynamic analysis is IDA.
• The modeling procedure doesn't have verification. Please verify modeling in OpenSees software.
• Please explain about the design procedure.
• The earthquake records should be introduced completely for example fault distance, PGA, PGV and essential outputs.
• Figure 9 shows IDA curves for 30 records but Figure 10 indicates IDA curves for 20 records. Please explain about this difference.
• Please explain the criterion of collapse in IDA. IDA curves don't show the collapse state.
• Please present fragility curves for LS performance level.
•The limitations of this research should be presented in the manuscript.
• The recommendation section should be added to the manuscript.
• The authors are recommended to seek the services of a professional copywriter or native speaker for thorough corrections. Some sentences are so long and it is difficult to understand.
Author Response

(The authors gave the same response as above.)

Round 2
Reviewer 1 Report
The paper was revised accordingly and the recommandation is to be published.
Reviewer 2 Report
please refer to the file of 'buildings-2640770 Comments2nd02.pdf'.

Reviewer 3 Report
Authors have provided acceptable answers to reviewer's questions.
Reviewer 4 Report
Dear Authors;
Thank you for your efforts to prepare this paper.